# Motivations, perceived risk, and tampering of XTAMPZA ER and abuse-deterrent opioid drugs

**Hannah L. Burkett** [ID]°\*, **Richard C. Dart**°, **Joshua C. Black**°

Rocky Mountain Poison & Drug Safety, Denver, Colorado, United States of America

☉ These authors contributed equally to this work.
\* hannah.burkett@rmpds.org

## Abstract

Abuse deterrent formulation (ADF) products are designed to prevent people from tampering with medication, especially with the intention to misuse or abuse the product. While clinical results support the efficacy of ADF products in controlled settings, little is known about how people use and abuse them in the real world. The objective of this research was to describe tampering behavior and perceived risk among U.S. adults who report non-medical use (NMU) of XTAMPZA® ER, a novel ADF extended-release opioid, and comparable products. A cross-sectional survey was conducted in 2021 and 2022 among adults who reported NMU of oxycodone or hydrocodone products in the past year. To compare use patterns and tampering behavior to XTAMPZA® ER, three similar opioid products were also assessed. Participants who reported NMU of an eligible drug were asked why they used it, how they tampered with it, and what other drug-related behaviors they engaged in. A total of 628 participants were recruited. A total of 56 (8.92%) reported NMU of XTAMPZA® ER in the past year, 256 (40.8%) reported NMU of other ADF ER oxycodone/hydrocodone, 459 (73.1%) reported NMU of non-ADF ER oxycodone/hydrocodone, and 225 (35.8%) reported NMU of IR oxycodone. Non-ADF ER oxycodone/hydrocodone was the most used. Those who reported NMU of XTAMPZA® ER had more severe drug use profiles, including higher scores on the 10-item Drug Abuse Screening Test (DAST-10) and higher prevalence of concomitant and illicit drug use. Overall, 3.7% tampered with XTAMPZA® ER, 12.9% tampered with other ADF ER oxycodone/hydrocodone, 16.0% tampered with non-ADF ER oxycodone/hydrocodone, and 10.7% tampered with IR oxycodone. U.S. adults who reported tampering with XTAMPZA® ER or a comparable product frequently did so for therapeutic reasons, most commonly "to swallow the pill more easily" and "to improve the pain relief from the pill." They also viewed tampering with products they used as less risky than tampering with products they did not use. Tampering with XTAMPZA® ER was uncommon compared to other drugs but more common as a proportion among those who used XTAMPZA® ER. This is, potentially because those using XTAMPZA® ER had other markers of severe, problematic drug use.

**Data availability statement:** All relevant data are within the manuscript and its Supporting Information files.

**Funding:** This work was funded by Collegium Pharmaceuticals, Inc. The funder did not participate in data collection, analysis, or interpretation of the findings.

**Competing interests:** The authors have declared that no competing interests exist.

## Introduction

It is estimated that 9.3 million people in the U.S. misused prescription pain relievers in 2020, [1] and opioid prescriptions per capita remains high [2]. Although immediate-release (IR) opioids continue to be diverted and abused at higher rates than extended-release (ER) formulations, among respondents in national surveys who report misusing opioids, over 40% report misusing crush-resistant ER formulations [3]. Tampering via physical and chemical manipulation or use through unintended routes can bypass the time release mechanism, resulting in the entire opioid load releasing and absorbing rapidly [4]. The risk of negative outcomes associated with abuse increase when the drug is not used as intended [5]. In 2017, the relative risk of death or major medical outcome compared to abuse by oral route was found to be 2.24 times for inhalation, 2.6 times for injection and 2.41 times for other/multiple routes of abuse [5]. Public health efforts to address tampering and abuse of ER opioid products have evolved in response to these risks.

Reformulating high risk drugs into abuse deterrent formulations (ADF) [6] is one of many interventions [7–9] deployed to curb overdose and related harms. ADFs are intended to make tampering harder and less rewarding, ensuring the time release load enters the system as intended. While most individuals with chronic pain do not abuse drugs [7], ADF products are designed to protect against tampering for therapeutic reasons, either intentional (e.g., To enhance pain relief) and unintentional (e.g., Breaking up large tablets to make them easier to swallow) [8,9]. Previous research of an abuse-deterrent product has shown that when products are reformulated, rates of abuse, diversion, death, and doctor shopping decreased and remained low. [10]. There are several ADF products currently on the market including XTAMPZA® ER, OxyContin®, Hysingla® ER, and multiple generic oxycodone products. But as ADF technology improves, so does the response to it.

Because tampering methods continue to evolve, increasingly complex deterrence mechanisms are needed, such as those employed in XTAMPZA® ER. XTAMPZA® ER is an ER ADF oxycodone product that uses DETERx® technology designed to reduce manipulation, first marketed in 2016. XTAMPZA® ER contains pharmaceutically active microspheres delivered in a capsule for oral administration. These microspheres have been shown to maintain their integrity after attempted manipulation using common household tools and chewing [11]. XTAMPZA® ER was granted abuse-deterrent labeling with respect to oral, nasal, and intravenous routes of administration based on pre-clinical and clinical evidence showing reduced likelihood of tampering compared to similar drugs [12]. Rates of abuse, misuse, diversion, and tampering of XTAMPZA® ER are low compared to IR oxycodone, other ADF ER opioids, and non-ADF ER opioids [10].

Although clinical evidence shows XTAMPZA® ER puts up barriers to tampering [8,9,13], relatively little is known about how effectively XTAMPZA® ER DETERx technology prevents tampering in a community setting, or how people perceive the risk of NMU of XTAMPZA® ER and comparable drugs. Interventions intended to prevent tampering and reduce the risks associated with drug use in the general population should be assessed for efficacy under real-world conditions. The U.S. Food and Drug

Administration (FDA) requires real-world evidence conducted in a post-market setting to measure true efficacy of interventions. Final guidelines for ADF product labeling released by the FDA in 2015 [14] include specific requirements to surveil real-world use of the drug product to assess the effectiveness of ADF technology in the community.

Exploring the real-world efficacy of ADF technology and how the technology may be overcome through a targeted population of non-medical users fulfills the FDA requirement for real-world, post-marketing surveillance. It will also help inform risk reduction strategies and prescribing guidelines, as well as provide insight into this high-risk population. However, identifying cases of NMU in the community setting is challenging and often requires targeted sampling techniques. These challenges are compounded by low drug volume [15] and low exposure prevalence [16] to drugs designed to deter abuse among pain patients.

The objective of this research was to describe tampering behavior, including motivations, methods, success, and perceived risk among U.S. adults who report NMU of a novel ADF opioid and comparable products. Those who use prescription opioids non-medically represent a high-risk group that is hard to access outside of a clinical setting. Additionally, we characterized the population of adults reporting NMU in terms of demographics, health measures, and indicators of problematic drug behavior. We used a targeted case-finding method from a general population sample, which enabled investigation into low prevalence behaviors involving an infrequently dispensed drug by individuals who may not interact with the healthcare system. Individuals who non-medically use XTAMPZA ER and a control group of comparable products were recruited from a larger, nationally representative survey, the Survey of Non-Medical Use of Prescription Drugs (NMURx). We compared motivations for NMU, tampering behaviors, and risk perception of XTAMPZA ER with other ER ADF opioids, non-ADF ER opioids, and IR oxycodone. We also compared the difficulty and effectiveness of different tampering methods commonly used in the real-world setting.

## Materials and methods

NMURx is a cross-sectional surveillance survey administered biannually to approximately 30,000 adults, which has been shown to be valid [17] and reliable [15]. The routine NMURx survey collects anonymous information about prescription drug use and associated behaviors. Respondents reporting past year NMU, defined as use "in a way not directed by a healthcare professional" of XTAMPZA ER or a comparable product were recruited into a follow-up survey sent within 2 weeks of their completion of the routine survey. Comparable products were classified as 1) other ADF ER oxycodone/hydrocodone products, 2) non-ADF ER oxycodone/hydrocodone products, or 3) IR oxycodone products. In addition to questions around drug use behavior, the routine NMURx survey covers participant demographics and socioeconomic characteristics and includes several validated screening instruments, including the DAST-10.

The DAST-10 is a validated screening instrument for problematic drug use. A score of 3 or larger on the continuous scale is a suitable indicator for risk of substance use disorder [18]. Inclusion of the DAST-10 and past year use of illicit substances allows for the comparison of relevant risk factors which may affect an individual's chances of tampering. Participants were recruited from the 3rd quarter 2021 and the 1st quarter 2022 routine surveys. The NMURx Program study protocol is approved by the Colorado Multiple Institutional Review Board, with the most recent certificate of exemption granted 17 January 2019.

The follow-up survey asked specific questions about past year NMU and tampering for XTAMPZA ER and its comparable products. Comparable products were selected based on indication and form. A full list of products included in each comparison group can be found in S1 Table. Respondents were first required to confirm they used one of the products non-medically in the past year and were then asked which product they preferred. Respondents were asked about specific potential tampering methods for each endorsed product and were then asked a series of follow-up questions for each endorsed method. Of the eight methods asked about, chewing, crushing, heating or melting, dissolving orally, and dissolving in a liquid were classified as tampering with the pill.

Follow up questions covered motivations, difficulty level (as a Likert scale), and whether the tampering was perceived as effective. Because ADF products are only intended for oral use, manipulating the product for use by another route was considered motivation. Difficulty level of tampering was averaged across all individuals who used each drug. Finally, respondents rated their perceived risk of harm by NMU of each product and their perceived risk of harm by engaging in seven use behaviors. on a series of 4-point Likert scales from no risk to great risk. Perceived risk was averaged both across all individuals and across only individuals who reported use of each drug to examine whether risk differs by history of use. All analyses were conducted in SAS 9.4 (Cary, NC). This work was funded by Collegium® Pharmaceuticals, Inc. The funder did not participate in data collection, analysis, or interpretation of the findings.

## Results

### Descriptive analysis

A total of 9,158 of respondents to the routine NMURx survey reported use of XTAMPZA ER, other ADF ER oxycodone/hydrocodone, non-ADF oxycodone/hydrocodone or IR oxycodone within the past 12 months and were invited to initiate the follow-up survey. Of those invited, 2,186 initiated the follow-up survey and 738 were eligible to complete the follow-up survey based on endorsing past year NMU of one or more qualifying products. Of these, 628 (85.1%) respondents completed the survey (Fig 1). Demographics for the final sample are shown in Table 1. The sample was predominantly female (52.7%), White (85.5%), non-Hispanic (89.3%), and had a DAST-10 level of low or none (82.6%). Of the targeted sample who reported NMU of at least 1 product, 56 (8.9%) endorsed NMU of XTAMPZA ER in the past year, 256 (40.0%) endorsed NMU of another ADF ER oxycodone/hydrocodone, 459 (73.1%) endorsed NMU of a non-ADF ER oxycodone/hydrocodone and 225 (35.0%) endorsed NMU of an IR oxycodone pain reliever in the past year.

Compared to XTAMPZA ER, other ADF ER oxycodone/hydrocodone products were dispensed at 42.2%, non-ADF ER oxycodone/hydrocodone were dispensed at 1,976.8% and IR oxycodone pain relievers were dispensed at 1.4% the

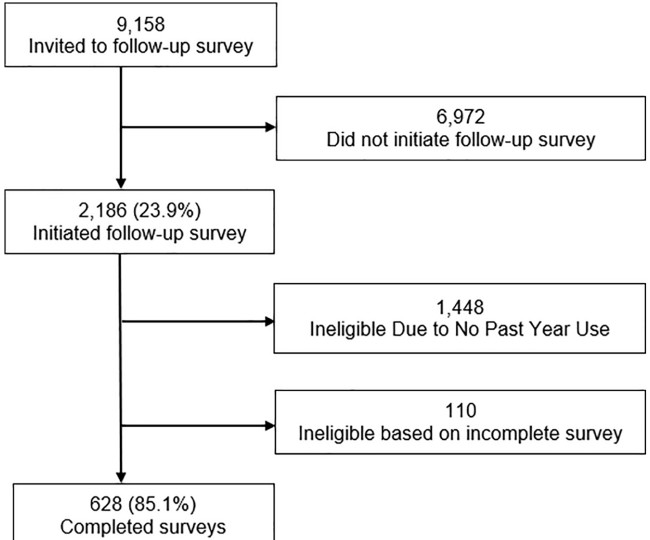

**Fig 1. Flow Chart of Survey Respondents.** The number and percent of respondents who initiated and completed the follow up survey out of those that were invited.

**Table 1. Demographics among adults who non-medically use by drug group in the past year.**

| Characteristic | Full Sample N (%) | XTAMPZA ER N (%) | ADF ER oxycodone or hydrocodone N (%) | Non-ADF ER oxycodone or hydrocodone N (%) | IR oxycodone N (%) |
|---|---|---|---|---|---|
| **Number of Participants** | 628 (100) | 56 (8.9) | 256 (40.8) | 459 (73.1) | 225 (35.8) |
| **Sex** | | | | | |
| Male | 297 (47.3) | 32 (57.1) | 131 (51.2) | 213 (46.4) | 113 (50.2) |
| Female | 331 (52.7) | 24 (42.9) | 125 (48.8) | 246 (53.6) | 112 (49.8) |
| **Age** | | | | | |
| 18–24 | 7 (1.1) | Suppressed | 5 (2.0) | 5 (1.1) | Suppressed |
| 25–34 | 67 (10.7) | 9 (16.1) | 31 (12.1) | 53 (11.5) | 24 (10.7) |
| 35–44 | 142 (22.6) | 23 (41.1) | 72 (28.1) | 94 (20.5) | 60 (26.7) |
| 45–54 | 102 (16.2) | 8 (14.3) | 42 (16.4) | 72 (15.7) | Suppressed |
| 55–64 | 143 (22.8) | 9 (16.1) | 48 (18.8) | 111 (24.2) | 43 (19.1) |
| 65 or Older | 167 (26.6) | Suppressed | 58 (22.7) | 124 (27.0) | 59 (26.2) |
| **Race[a]** | | | | | |
| Black or African American | 56 (8.9) | Suppressed | 22 (8.6) | 41 (8.9) | 19 (8.4) |
| White | 537 (85.5) | 47 (83.9) | 215 (84.0) | 391 (85.2) | 188 (83.6) |
| Other | 47 (7.5) | Suppressed | 23 (9.0) | 38 (8.3) | 22 (9.8) |
| **Ethnicity** | | | | | |
| Hispanic or Latino | 67 (10.7) | 7 (12.5) | 34 (13.3) | 46 (10.0) | 29 (12.9) |
| **Household Income** | | | | | |
| Less than $25,000 | 126 (20.1) | 11 (19.6) | 47 (18.4) | 92 (20.0) | 53 (23.6) |
| Between $25,000 and $49,999 | 163 (26.0) | 13 (23.2) | 59 (23.0) | 130 (28.3) | 54 (24.0) |
| Between $50,000 and $74,999 | 116 (18.5) | Suppressed | 49 (19.1) | 84 (18.3) | 40 (17.8) |
| Between $75,000 and $99,999 | 92 (14.6) | Suppressed | 40 (15.6) | 62 (13.5) | 36 (16.0) |
| $100,000 or more | 131 (20.9) | 20 (35.7) | 61 (23.8) | 91 (19.8) | 42 (18.7) |
| **Current Student Status** | | | | | |
| Yes | 41 (6.5) | 8 (14.3) | 25 (9.8) | 28 (6.1) | 15 (6.7) |
| **Military Service** | | | | | |
| Yes | 79 (12.6) | 11 (19.6) | 41 (16.0) | 55 (12.0) | 29 (12.9) |
| **DAST-10 Level** | | | | | |
| Low level or none reported (0–2) | 519 (82.6) | 36 (64.3) | 202 (78.9) | 375 (81.7) | 178 (79.1) |
| Moderate to severe level (3–10) | 109 (17.4) | 20 (35.7) | 54 (21.1) | 84 (18.3) | 47 (20.9) |
| **Use of Illicit Drugs in Past Year** | | | | | |
| Yes | 109 (17.4) | 18 (32.1) | 58 (22.7) | 77 (16.8) | 47 (20.9) |
| **Preferred Drug Used in Past Year** | | | | | |
| Xtampza ER | 16 (2.5) | 16 (28.6) | 7 (2.7) | 7 (1.5) | 8 (3.6) |
| ADF ER pain relievers | 138 (22.0) | 13 (23.2) | 138 (53.9) | 61 (13.3) | 42 (18.7) |
| Non-ADF ER pain relievers | 281 (44.7) | 14 (25.0) | 44 (17.2) | 281 (61.2) | 53 (23.6) |
| IR Oxycodone pain relievers | 97 (15.4) | 5 (8.9) | 32 (12.5) | 46 (10.0) | 97 (43.1) |
| Other prescription pain reliever | 96 (15.3) | 8 (14.3) | 35 (13.7) | 64 (13.9) | 25 (11.1) |

*Respondents can endorse more than one drug group, so sample sizes from the drug groups will not add up to the full sample.*

average number of units per year-quarter. Those who endorsed NMU of XTAMPZA ER were younger (mean age = 44.2) than those who non-medically used other drugs (mean age = 53.9). Overall, 16 respondents (2.6%) reported XTAMPZA ER as their preferred drug, while 281 (44.8%) preferred another non-ADF ER oxycodone/hydrocodone.

## Tampering methods

A total of 168 (26.8%) of respondents reported tampering with one of the drugs in the past year. Of those reporting NMU of each product, 23 (41.1%, N = 56) reported tampering with XTAMPZA ER, 81 (31.6%, N = 256) tampered with another ADF ER oxycodone/hydrocodone, 103 (22.4%, N = 40.8) tampered with a non-ADF ER oxycodone/hydrocodone, and 67 (29.8%, N = 225) tampered with IR oxycodone. Among those who reported NMU, the percentages of tampering by different methods were generally similar across drug groups (Table 2). Among those who tampered with XTAMPZA ER, 14 (60.9%) dissolved it orally, 14 (60.9%) dissolved it into a liquid, 10 (43.5%) chewed it, and 9 (39.1%) crushed it. For other ADF ER oxycodone/hydrocodone and IR oxycodone, crushing was the most common method; for non-ADF oxycodone/hydrocodone, dissolving the pill orally was most common. The least common method for all products was heating or melting, although this method was more common with ADF ER oxycodone/hydrocodone or XTAMPZA ER than non-ADF oxycodone/hydrocodone or IR oxycodone.

Reasons for tampering with products were relatively consistent across the drug groups, with the most common reasons being to increase the high feeling from the pill, to improve pain relief from the pill, to swallow the pill more easily, and to feel the effect of the pill more quickly. For those who reported tampering with XTAMPZA ER, 87.0% did so to feel the effects of the pill more quickly, 73.9% did so to improve pain relief from the pill, and 73.9% did so to swallow the pill more easily. Among respondents who reported tampering with any product, most reported that the method they used was successful for feeling the effects of the pill more quickly, improving pain relief from the pill, and swallowing the pill more easily.

**Table 2. Tampering and behavioral risk factors by drug group.**

| Characteristic | Full Sample N (%) | XTAMPZA ER N (%) | ADF ER oxycodone or hydrocodone N (%) | Non-ADF ER oxycodone or hydrocodone N (%) | IR oxycodone N (%) |
|---|---|---|---|---|---|
| Number of Participants who Tampered with Each Drug | 168 (26.8) | 23 (41.1) | 81 (31.6) | 103 (22.4) | 67 (29.8) |
| **Method of tampering** | | | | | |
| Chewed | 81 (46.8) | 10 (43.5) | 34 (42.0) | 43 (41.7) | 31 (46.3) |
| Crushed | 90 (52.0) | 9 (39.1) | 46 (56.8) | 50 (48.5) | 36 (53.7) |
| Heated or melted | 49 (28.3) | 8 (34.8) | 28 (34.6) | 25 (24.3) | 16 (23.9) |
| Dissolved Orally | 103 (59.5) | 14 (60.9) | 39 (48.1) | 59 (57.3) | 33 (49.3) |
| Dissolved into a liquid | 67 (38.7) | 14 (60.9) | 31 (38.3) | 31 (30.1) | 24 (35.8) |
| **Reason for Tampering** | | | | | |
| To swallow the pill more easily | 128 (74.0) | 17 (73.9) | 60 (74.1) | 77 (74.8) | 46 (68.7) |
| To inject the contents of the pill | 64 (37.0) | 12 (52.2) | 38 (46.9) | 34 (33.0) | 24 (35.8) |
| To smoke or vape the contents of the pill | 56 (32.4) | 13 (56.5) | 28 (34.6) | 30 (29.1) | 21 (31.3) |
| To snort the contents of the pill | 76 (43.9) | 16 (69.6) | 35 (43.2) | 42 (40.8) | 32 (47.8) |
| To increase the high feeling from the pill | 97 (56.1) | 19 (82.6) | 53 (65.4) | 54 (52.4) | 34 (50.7) |
| To improve the pain relief from the pill | 127 (73.4) | 17 (73.9) | 59 (72.8) | 76 (73.8) | 49 (73.1) |
| To feel the effects of the pill more quickly | 121 (69.9) | 20 (87.0) | 58 (71.6) | 73 (70.9) | 46 (68.7) |
| For another reason | 56 (32.4) | 11 (47.8) | 23 (28.4) | 26 (25.2) | 22 (32.8) |
| **Outcomes** | | | | | |
| ER visits | 57 (8.8) | 16 (28.6) | 30 (11.7) | 32 (7.0) | 20 (8.9) |
| Naloxone | 62 (9.6) | 16 (28.6) | 29 (11.3) | 36 (7.8) | 25 (11.1) |
| **Risk Behaviors** | | | | | |
| Taking more than recommended | 175 (27.1) | 23 (41.1) | 69 (27.0) | 108 (23.5) | 56 (24.9) |
| Concomitant use with a sedative | 157 (24.3) | 22 (39.3) | 58 (22.7) | 108 (23.5) | 58 (25.8) |
| Concomitant use with alcohol | 125 (19.4) | 17 (30.4) | 58 (22.7) | 86 (18.7) | 38 (16.9) |

Most tampering methods were ranked on average, as moderately difficult. Those who reported tampering with XTAMPZA ER reported that dissolving it in their mouth was easier than other methods, which is not the case for other ER oxycodone/hydrocodone. Compared to other products, XTAMPZA ER was reported as harder to chew and dissolve into a liquid but easier to heat or melt (Fig 2).

### Perceived risk

Among the full sample, the perceived NMU risk of XTAMPZA ER and other ADF ER oxycodone/hydrocodone was lower than the perceived NMU risk of non-ADF ER oxycodone/hydrocodone and IR oxycodone (Fig 3). When referring to drugs that respondents themselves had non-medically used, the perceived NMU risk of XTAMPZA ER was higher, and only the perceived NMU risk of ADF ER oxycodone/hydrocodone was lower. The perceived risk of tampering with XTAMPZA ER and other ADF ER oxycodone/hydrocodone is higher among those who reported tampering with that drug in the past year compared to those who did not report tampering with that drug. For non-ADF ER oxycodone/hydrocodone and IR oxycodone, the perceived risk of tampering is similar between those who did and did not report tampering with that drug.

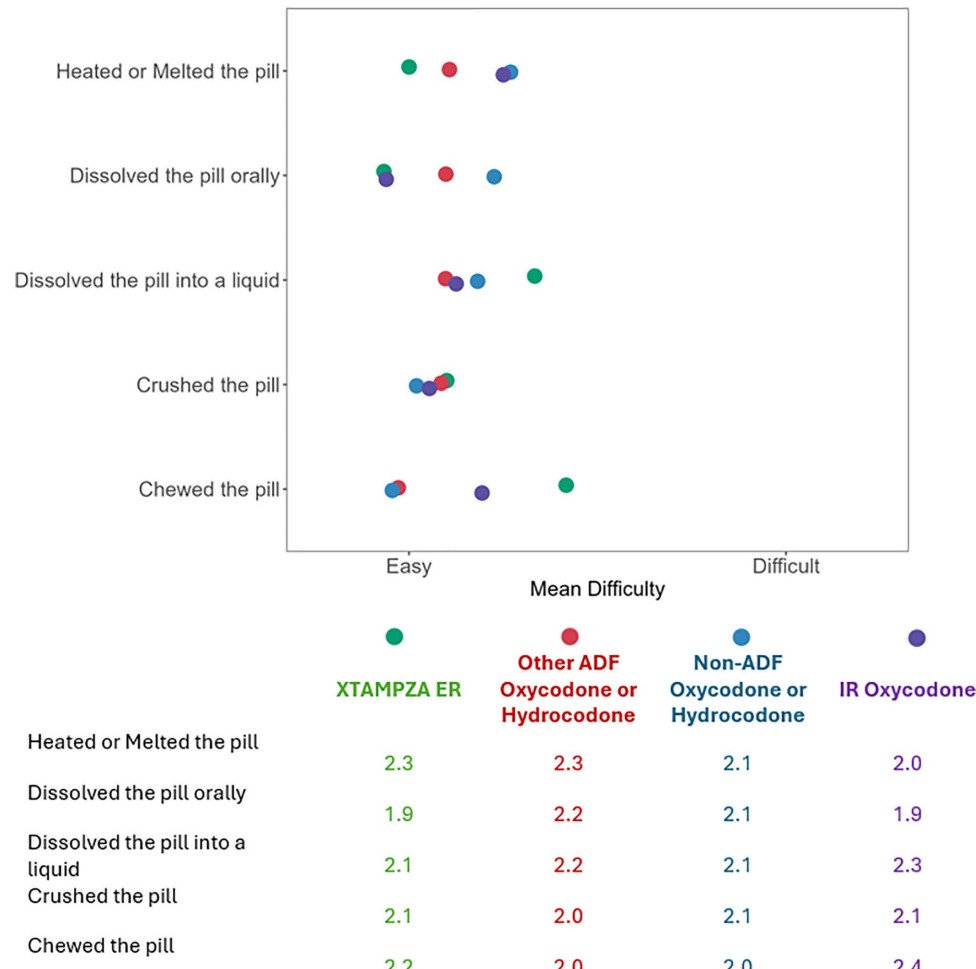

| | XTAMPZA ER | Other ADF Oxycodone or Hydrocodone | Non-ADF Oxycodone or Hydrocodone | IR Oxycodone |
|---|---|---|---|---|
| Heated or Melted the pill | 2.3 | 2.3 | 2.1 | 2.0 |
| Dissolved the pill orally | 1.9 | 2.2 | 2.1 | 1.9 |
| Dissolved the pill into a liquid | 2.1 | 2.2 | 2.1 | 2.3 |
| Crushed the pill | 2.1 | 2.0 | 2.1 | 2.1 |
| Chewed the pill | 2.2 | 2.0 | 2.0 | 2.4 |

**Fig 2. Reported difficulty of tampering by method.** Mean reported difficulty of tampering with each drug group by each individual method among respondents who reported tampering with each drug by each individual method.

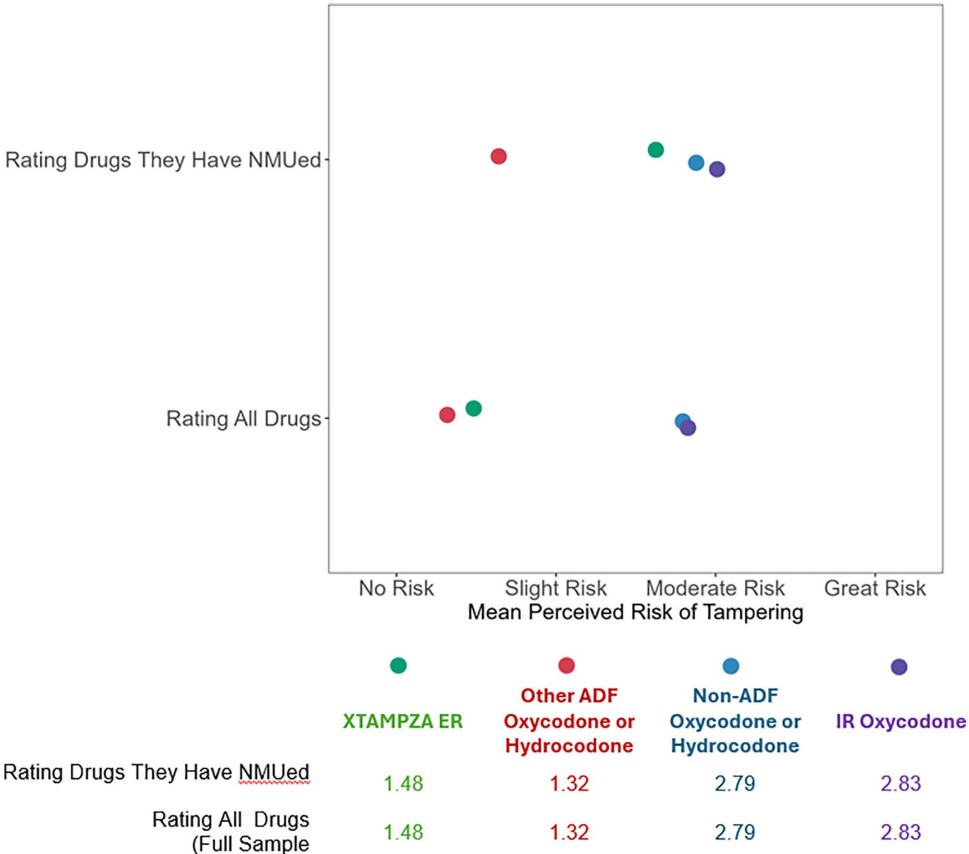

| | XTAMPZA ER | Other ADF Oxycodone or Hydrocodone | Non-ADF Oxycodone or Hydrocodone | IR Oxycodone |
|---|---|---|---|---|
| Rating Drugs They Have NMUed | 1.48 | 1.32 | 2.79 | 2.83 |
| Rating All Drugs (Full Sample | 1.48 | 1.32 | 2.79 | 2.83 |

**Fig 3. Perceived Risk of Tampering by drug group.** Mean reported perceived risk of tampering with each drug group among the full sample and among only those who reported NMU of each drug.

## Discussion

Even among a targeted sample of adults who reported use of one or more qualifying products in the past year, only 33.8% reported past year NMU and only 7.7% reported tampering. The low prevalence of these behaviors makes them difficult to quantify and describe without using a targeted sampling approach. Evidence generated in a real-world setting has become increasingly recognized as an essential part of post-market drug surveillance [19]. FDA recently produced a draft guidance on the use of real-world evidence in regulatory decision-making, where conclusions from such studies could be used to inform healthcare and regulatory decisions [19]. Obtaining real world evidence from a community setting is more challenging when it requires evaluation of low prevalence behaviors. Those who report NMU of drugs and tampering are the type of patient ADF technology is intended for, and targeted sampling is required to recruit these individuals from a community setting.

NMU of XTAMPZA ER was uncommon relative to similarly indicated oxycodone- and hydrocodone-containing drugs. Compared to the full sample, those who reported NMU of XTAMPZA ER also reported more severe drug use profiles (e.g., higher DAST-10 score, concomitant drug use, illicit drug use) than those using comparator drugs. This could be due to differential prescribing practices between XTAMPZA ER and other ADF products. A previous study found that 80% of physicians prescribing an ADF were influenced in their decision to prevent patients from switching to heroin [20], suggesting that existing abuse behaviors may be the driving factor in ADF prescriptions.

The preferred drug for past year use (including medical and non-medical) among adults reporting NMU was non-ADF ER pain relievers with less than 3.0% preferring XTAMPZA ER. Participants' preferred drug tended to be the same drug they used non-medically. A similar pattern was seen in perceived risk of tampering among ADF products. Respondents rated the risk higher among ADF drugs they used non-medically compared to those they had not. This was not the case for non-ADF products where the perceived risk remained stable between those who had and had not engaged in NMU behavior with each product.

Among non-medical users who reported tampering, the proportion that tampered with XTAMPZA ER was lower than for all comparator drug groups. Among the small group who reported NMU of each drug group, a greater proportion reported tampering with XTAMPZA ER compared to other drugs. Among those who reported tampering with XTAMPZA ER, the majority manipulated the pill in ways that would likely not disrupt the ADF properties [8,9,13], decreasing the risk of harmful outcomes normally associated with drug tampering.

The most common reason reported for tampering with any qualifying product was to swallow the pill more easily. This may be motivated by therapeutic reasons and patients may not know that tampering to achieve these goals can disrupt the time release mechanism and put them at higher risk of harmful outcomes [5]. In these well intended situations, crushing or breaking pills may occur in therapeutic settings [21] or may be done for therapeutic reasons. Products that can be manipulated to be easier to swallow without disrupting the time release mechanism, such as XTAMPZA ER, could protect patients from unknowingly engaging in high-risk behaviors (S1 Table).

The study has several strengths, one being inclusion of an appropriate control group from the same population, which can be challenging when drug indication and prescribing practices affect an individual's exposure to specific products. Recruiting the sample from a general population also allowed a more comprehensive group of individuals to be analyzed, and for response bias among the final targeted sample to be measured. The outcomes of interest in this study were low prevalence, high risk behaviors which may not occur within clinical study samples. The survey was anonymously conducted, which could further reduce bias [22].

The study had two main limitations. The first was that information about prior medical history was not collected, and therefore some context could not be discerned. It cannot be discerned if the more severe drug use profiles seen among adults who reported NMU of XTAMPZA ER are due to differential prescribing for adults with a history of drug NMU. The second is that individuals who experienced severe harm from engaging in high-risk drug behavior may not be recruited by a self-report survey. Further, selection bias into the follow-up survey is unaccounted for, preventing strict statistical comparisons, although the 85% of those who began the follow-up survey and were found eligible did complete the survey.

By recruiting from a from a general population source survey shown to be broadly representative of national trends in drug use [17] and employing both careless response exclusions [23] and confirmatory answering [24], we were able to test for differences in medical outcomes or behavioral patterns in a case-control study design with confounding control, to provide relevant real-world data on use and misuse of ADF products required for appropriate evaluation of the technology. Our results suggest that a substantial percent of adults who tamper with abuse-deterrent drugs may be doing it for therapeutic reasons such as to swallow the pill more easily. Although this is still misuse, patients may not be aware that this carries the same risks associated with abusing abuse-deterrent formulations of oxycodone and hydrocodone. The ability to capture this high-risk group of adults engaging in low prevalence behaviors begins to provide insight into the real-world motivations for drug tampering.

## Supporting information

**S1 Table. Product Listing of Drug Comparator Groups.**
(DOCX)

**S1 File. Survey Wording.**
(DOCX)

**S2 File. Raw Data.**
(XLSX)

## Acknowledgments

Authors have no individual declarations of interest. This work was performed in collaboration with the Researched Abuse, Diversion and Addiction-Related Surveillance (RADARS®) System. The RADARS System is supported by subscriptions from pharmaceutical manufacturers, government, and non-government agencies for surveillance, research and reporting services. RADARS System is the property of Denver Health and Hospital Authority (DHHA), a political subdivision of the State of Colorado. Denver Health retains exclusive ownership of all data, databases and systems. This work was funded by Collegium Pharmaceuticals, Inc. The funder did not participate in data collection, analysis, or interpretation of the findings.

## Author contributions

**Conceptualization:** Joshua C Black.

**Formal analysis:** Hannah L Burkett.

**Funding acquisition:** Richard C Dart.

**Investigation:** Hannah L Burkett.

**Methodology:** Joshua C Black.

**Project administration:** Joshua C Black.

**Supervision:** Richard C Dart, Joshua C Black.

**Validation:** Joshua C Black.

**Visualization:** Hannah L Burkett.

**Writing – original draft:** Hannah L Burkett.

**Writing – review & editing:** Joshua C Black.

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
