## [Decision Letter · Decision Letter 0]

17 Dec 2024

Dear Dr. Burkett,

Thank you for submitting your manuscript to PLOS ONE. After careful consideration, we feel that it has merit but does not fully meet PLOS ONE’s publication criteria as it currently stands. Therefore, we invite you to submit a revised version of the manuscript that addresses the points raised during the review process.

**COMMENTS OF ACADEMIC EDITOR:**

We look forward to receiving your revised manuscript.

Kind regards,

Amisha Raikar, 

**Guest Editor**

**PLOS ONE**

4. Your abstract cannot contain citations. Please only include citations in the body text of the manuscript, and ensure that they remain in ascending numerical order on first mention.

Additional Editor Comments :

1. Revise the objective to succinctly state the purpose and method of the study in more accessible terms.

2. Provide a brief background on ADFs and their purpose before moving to specific examples like XTAMPZA ER.

3. While the DAST-10 threshold for substance use disorder is mentioned, include a brief justification for using this specific tool and how it contributes to the study outcomes.

4. Some sentences are very long and packed with multiple pieces of information. Break them into shorter, clearer sentences to improve readability.

5. Consider defining terms like "RADARS System" and "DAST-10" for readers who may not be familiar with these methodologies.

6. In the discussion section, the ideas are densely packed and occasionally jump between topics (e.g., tampering behaviors, prescribing practices, and study strengths). Consider organizing the discussion into distinct paragraphs for each main topic (e.g., prevalence of NMU, tampering motivations, prescribing practices, study strengths, and limitations).

6. Expand on the implications of using real-world evidence (RWE) in post-market surveillance and how this study aligns with FDA guidance. Consider tying this to broader trends in RWE uses for regulatory decision-making

Reviewers' comments:

Reviewer's Responses to Questions

**Comments to the Author**

1. Is the manuscript technically sound, and do the data support the conclusions?

Reviewer #1: No

Reviewer #2: Partly

2. Has the statistical analysis been performed appropriately and rigorously?

Reviewer #1: No

Reviewer #2: No

3. Have the authors made all data underlying the findings in their manuscript fully available?

Reviewer #1: Yes

Reviewer #2: Yes

4. Is the manuscript presented in an intelligible fashion and written in standard English?

Reviewer #1: Yes

Reviewer #2: Yes

Reviewer #1: Thank you for giving me the opportunity to review the manuscript. I read it with interest.

This study addresses an important public health problem in the U.S. regarding the prevention of misuse and harmful outcomes associated with non-medical use of opioids. The authors attempt to explore the usefulness of XTAMPZA ER through a survey-based approach. The Abuse-deterrent formulations (ADF) encouraged by the FDA could be effective in preventing the inappropriate use of opioids. The ADF can protect against the unintentional use of opioids to achieve a feeling of euphoria; it will avoid a sharp increase in blood opioid concentration level. As a consequence, the ADF would be considered to reduce the incidence of opioid abuse and acute poisoning. Examining the effectiveness of the ADF is crucial for national institutes developing policies against the opioid crisis, citizens exposed to the danger of harm from opioids, and the ADF opioid manufacturers. While the evidence of the ADF is sought, it is challenging to study the cases of non-medical use of opioids in the community, as the authors wrote.

The authors conducted an online survey using the RADARS System. However, I think this survey could not achieve the authors' aim because of the issue of the online panel. The respondents of this survey are limited to non-medical use opioid survivors. And vice versa, people who had died or experienced severe harm by the non-medical use of opioids could not answer this survey. Thus, I think that this study could not show the actual effect of ADF on non-medical use of opioids.

Moreover, the result that only 628 of 9158 respondents answered the follow-up survey suggests that the respondents might be biased toward those with high literacy. People who do not have high literacy may use and tamper more than respondents.

In the results section, the authors indicated that responders can endorse more than one drug. In other words, each drug group includes concomitant use of other formula opioids. Therefore, I assume that these results could not show the actual number of regarding characteristics, tampering, and other items in only XTAMPZA ER use.

I am also wondering why the authors did not consider the consumption of XTAMPZA ER and the opioids used as a comparator in this study.

For these reasons, I'm afraid I have to disagree with the author's conclusion: "Among similar drugs, XTAMPZA ER was not the drug of choice for non-medical use, and popular methods of tampering may not involve increased risk of harmful outcomes."

To better address the research objectives, I suggest that authors consider combining the results of several surveys derived from different data sources in future studies. Consider Mortality Records, Emergency Department Visit and Hospitalization data, Poison Center data, etc.

While the study addresses an important issue, the limitations in the methodology significantly constrain its ability to answer the research question. For these reasons, I recommend rejecting the manuscript in its current form. However, the study topic remains important, and I encourage the authors to consider revising their approach to better align with their research objectives.

Reviewer #2: The authors present an interesting study that warrants discussion. Given the ongoing opioid epidemic, the public health implications that studies of this nature explore are noteworthy. While the present study raises important questions regarding a public health crisis, I do have a few questions and suggestions.

There are multiple sentences that are confusing (e.g., sentence starting on line 73).

Introduction

• ER is clarified as “extended-release” in the first paragraph and does not need to be re-clarified again (e.g., start of the third paragraph).

• See previous comment for immediate release (IR).

• What is the nationally representative survey participants were recruited from?

Methods

• I am confused about what survey participants were pulled from. You mention RADARS and NMURx, but it is not clear how both surveys were utilized in this study.

• The first paragraph needs to be split into two.

• The description of each of the focal variables is difficult to discern since everything is mentioned in a single paragraph. There is no mention of the sociodemographic variables (mentioned in the results section).

Results

• I think there is a formatting issue with the sub-header lines for “Figure 1…” and for “Figure 2…”

Discussion

• You do not clarify what the abbreviation of FDA refers to, even though it is a commonly discussed agency in the US. You mention the agency’s full name earlier in the paper, but do not include the abbreviation.

**Do you want your identity to be public for this peer review?** For information about this choice, including consent withdrawal, please see our Privacy Policy

Reviewer #1: No

Reviewer #2: No

---

## [Author Response · Author response to Decision Letter 1]

28 Jan 2025

To the editors:

Thank you for your interest and the opportunity to revise our manuscript. We have responded to the reviewer comments below and uploaded a modified manuscript for consideration. We have also updated manuscript formatting based on the PLOS One style guide.

Editor Comment 1: Revise the objective to succinctly state the purpose and method of the study in more accessible terms.

AUTHOR RESPONSE EDITOR COMMENT 1: We have revised the objective and stated it plainly beginning on line 107.

Editor Comment 2: Provide a brief background on ADFs and their purpose before moving to specific examples like XTAMPZA ER.

AUTHOR RESPONSE EDITOR COMMENT 2: We added a sentence describing that ADF’s are an intervention to reduce the risks of ER opioid misuse by creating physical barriers to tampering beginning on line 56.

Editor Comment 3: While the DAST-10 threshold for substance use disorder is mentioned, include a brief justification for using this specific tool and how it contributes to the study outcomes.

AUTHOR RESPONSE EDITOR COMMENT 3: We added description of why we characterized the sample in terms of problematic drug use behavior beginning on line 110 and a description of the DAST-10 validated instrument beginning on line 153.

Editor Comment 4: Some sentences are very long and packed with multiple pieces of information. Break them into shorter, clearer sentences to improve readability.

AUTHOR RESPONSE EDITOR COMMENT 4: We edited the full manuscript and broke up sentences and paragraphs where necessary to improve readability.

Editor Comment 5: Consider defining terms like "RADARS System" and "DAST-10" for readers who may not be familiar with these methodologies.

AUTHOR RESPONSE EDITOR COMMENT 5: We removed mention of the RADARS System and added a description of the DAST-10 beginning on line 153. We reviewed the manuscript for additional terms which were possibly not well defined and added more complete definitions.

Editor Comment 6: In the discussion section, the ideas are densely packed and occasionally jump between topics (e.g., tampering behaviors, prescribing practices, and study strengths). Consider organizing the discussion into distinct paragraphs for each main topic (e.g., prevalence of NMU, tampering motivations, prescribing practices, study strengths, and limitations).

AUTHOR RESPONSE EDITOR COMMENT 6: We have reorganized the discussion section by main topics, ensuring multiple topics do not appear within the same paragraph.

Editor Comment 7: Expand on the implications of using real-world evidence (RWE) in post-market surveillance and how this study aligns with FDA guidance. Consider tying this to broader trends in RWE uses for regulatory decision-making

AUTHOR RESPONSE EDITOR COMMENT 7: We added a deeper discussion of the need for real-world evidence in post-marketing surveillance as well as a description of how our study meets those needs beginning on line 267.

Reviewer #1

Comment 1: The authors conducted an online survey using the RADARS System. However, I think this survey could not achieve the authors' aim because of the issue of the online panel. The respondents of this survey are limited to non-medical use opioid survivors. And vice versa, people who had died or experienced severe harm by the non-medical use of opioids could not answer this survey. Thus, I think that this study could not show the actual effect of ADF on non-medical use of opioids.

AUTHOR RESPONSE 1: Survivor bias is a limitation of this study as it would be with any data source that uses retrospective self-reported individual level data. I have added this as a limitation to the current study beginning on line 320.

Comment 2: Moreover, the result that only 628 of 9158 respondents answered the follow-up survey suggests that the respondents might be biased toward those with high literacy. People who do not have high literacy may use and tamper more than respondents.

AUTHOR RESPONSE 2: Of the 9,158 respondents who were invited to initiate the follow-up survey, 2,186 initiated the follow up survey, but only 738 were eligible to complete the survey based on past year reported NMU. This information is clarified beginning on line 189.

Comment 3: In the results section, the authors indicated that responders can endorse more than one drug. In other words, each drug group includes concomitant use of other formula opioids. Therefore, I assume that these results could not show the actual number of regarding characteristics, tampering, and other items in only XTAMPZA ER use.

AUTHOR RESPONSE 3: Respondents who endorsed more than one product were asked all relevant follow up questions separately in regards to each product. All reported metrics are specific to the products in each drug group presented.

Comment 4: I am also wondering why the authors did not consider the consumption of XTAMPZA ER and the opioids used as a comparator in this study.

AUTHOR RESPONSE 4: Consumption of XTAMPZA ER by several routes was considered and described in Table 2.

Comment 5: For these reasons, I'm afraid I have to disagree with the author's conclusion: "Among similar drugs, XTAMPZA ER was not the drug of choice for non-medical use, and popular methods of tampering may not involve increased risk of harmful outcomes."

AUTHOR RESPONSE 5: Among adults that reported NMU of a qualifying product, less than 3% reported XTAMPZA ER as their preferred drug. We have added clarification in the results and discussion sections that all conclusions are specific to the sample of adults reporting NMU.

Comment 6: To better address the research objectives, I suggest that authors consider combining the results of several surveys derived from different data sources in future studies. Consider Mortality Records, Emergency Department Visit and Hospitalization data, Poison Center data, etc.

AUTHOR RESPONSE 6: The objective of this work is to characterize how adults who report NMU of ADF products and their comparators use and tamper with these products in a community setting. This requires self-reported data sampled from the general population to capture adults who do not interact with the healthcare system. The goal of this work is to compliment the clinical studies that have been conducted. Such a study using clinical data has already been published by our group: https://doi.org/10.1093/pm/pnaa272. This work expands upon the past work by adding information from those in a community setting.

Comment 7: While the study addresses an important issue, the limitations in the methodology significantly constrain its ability to answer the research question. For these reasons, I recommend rejecting the manuscript in its current form. However, the study topic remains important, and I encourage the authors to consider revising their approach to better align with their research objectives.

AUTHOR RESPONSE 7: The research objective has been more clearly stated beginning on line 101. The manuscript has been reframed to more directly meet those objectives.

Reviewer #2

Comment 1: There are multiple sentences that are confusing (e.g., sentence starting on line 73).

AUTHOR RESPONSE 1: We have edited the full manuscript and updated any sentences that were confusing.

Comment 2: ER is clarified as “extended-release” in the first paragraph and does not need to be re-clarified again (e.g., start of the third paragraph).

AUTHOR RESPONSE 2: We removed the clarification.

Comment 3: See previous comment for immediate release (IR).

AUTHOR RESPONSE 3: We removed the clarification.

Comment 4: What is the nationally representative survey participants were recruited from?

AUTHOR RESPONSE 4: The nationally representative survey is the Survey of Non-Medical Use of Prescription Drugs (NMURx) routine survey. We added this clarification beginning on line 117.

Comment 5: I am confused about what survey participants were pulled from. You mention RADARS and NMURx, but it is not clear how both surveys were utilized in this study.

AUTHOR RESPONSE 5: Survey participants for the follow-up survey were recruited from the NMURx routine survey which is launched 2 times annually as part of the RADARS system. We have removed reference to RADARS as it is not directly relevant and reworked the methods section to clearly describe the two NMURx surveys and how they were each used for this research. Past validation studies for the NMURx survey are cited.

Comment 6: The first paragraph [of the methods] needs to be split into two.

AUTHOR RESPONSE 6: We split the first paragraph of the methods into one paragraph describing the routine NMURx survey and a second paragraph describing the follow-up survey.

Comment 7: The description of each of the focal variables is difficult to discern since everything is mentioned in a single paragraph.

AUTHOR RESPONSE 7: We have reorganized the methods section to describe each of the variables one at a time split thoughtfully into 3 paragraphs.

Comment 8: There is no mention of the sociodemographic variables (mentioned in the results section).

AUTHOR RESPONSE 8: We added a description of the sociodemographic variables taken from the routine NMURx survey beginning on line 136.

Comment 9: I think there is a formatting issue with the sub-header lines for “Figure 1…” and for “Figure 2…”

AUTHOR RESPONSE 9: We have updated the sub-headers to both figures and addressed the formatting errors.

Comment 9: You do not clarify what the abbreviation of FDA refers to, even though it is a commonly discussed agency in the US. You mention the agency’s full name earlier in the paper, but do not include the abbreviation.

AUTHOR RESPONSE 9: We added a clarifying abbreviation following the first mention of the agency’s full name on line 91.

---

## [Decision Letter · Decision Letter 1]

20 Mar 2025

Dear Dr. Burkett,

Thank you for submitting your manuscript to PLOS ONE. After careful consideration, we feel that it has merit but does not fully meet PLOS ONE’s publication criteria as it currently stands. Therefore, we invite you to submit a revised version of the manuscript that addresses the points raised during the review process.

We look forward to receiving your revised manuscript.

Kind regards,

Amisha Raikar, Master of Pharmaceutics

Guest Editor

PLOS ONE

Journal Requirements:

**Comments to the Author**

Reviewer #1: (No Response)

Reviewer #2: All comments have been addressed

2. Is the manuscript technically sound, and do the data support the conclusions?

Reviewer #1: Partly

Reviewer #2: Yes

3. Has the statistical analysis been performed appropriately and rigorously?

Reviewer #1: N/A

Reviewer #2: Yes

4. Have the authors made all data underlying the findings in their manuscript fully available?

Reviewer #1: Yes

Reviewer #2: Yes

5. Is the manuscript presented in an intelligible fashion and written in standard English?

Reviewer #1: Yes

Reviewer #2: No

Reviewer #1: Thank you for answering my comments and revising your manuscript. I have additional comments to improve your manuscript to make it better.

Abstract

(1) Line 18: I could not understand the meaning of the term "drug use profiles" until I read the Discussion(line 227). I recommend that the authors add some explanation.

(2) Line 20: I guess 16� is "non-"ADF ER.

(3) Are decimal points in percentages rounded down? Is it rounded off? It does not seem to be consistent.

Materials and Methods

(1) Even if the authors wrote about the sponsor of this research in Acknowledgment, I think that it should also be written in Materials and Methods.

(2) Line 106-� I understand the classification of comparable products, however, I could not know how the authors chose comparable products. Additionally, the list of products should be shown in a table.

(3) Lines 30-43: The authors wrote the number of participants. It should also be shown as a flow diagram.

(4) The questionnaire of this survey should be shown.

Result

(1) Table 2: In the "Reason for Tampering", "To increase the high feeling from the pill", "To improve the pain relief from the pill", and "To feel the effects of the pill more quickly" should be separate from other items. I think that "To swallow", "inject", "smoke or vape", and "snort" mean only change the route, do not mean the reasons. These are different concepts.

(2) Table 2: In the "Risk Behaviors and Outcomes" should be distinguished from "Risk Behaviors" and "Outcomes".

(3) Figures should be improved. I can imagine what the authors want to show in the Figures, however, it is unclear on what basis each point is placed in its respective position. I know the two X axes show "mean difficulty" and "mean perceived risk of tampering", respectively. The actual value of the respective means should also be specified somewhere.

Discussion

(1) Lines 214: The figures 33.8� and 7.7% are not shown in the Result section and Tables. It is unfriendly for readers. I recommend that the authors show these in the Results section.

(2) The number of consumptions, the number of prescriptions dispensed, or the ratio of the products surveyed in this study in the US is required. If XTAMPZA is not commonly prescribed and distributed in the US, the number of participants who non-medically use XTAMPZA will also be small.

Reviewer #2: I appreciate the author’s response to my comments. I have some additional suggestions listed below.

Please double-check punctions and grammar.

Introduction

• There is an extra “.” On line 41.

• Need to capitalize the start of the sentence on line 74.

• Missing a period at the end of the sentence on line 76.

• There is an extra “.” and “,” on line 80.

• You explain what NMU stands for in line 95, but it needs to occur at the first usage of NMU (line 85).

Results

• This may be an oversight on my end, but I do not see where Figures 1 or 2 are supposed to be placed in the manuscript.

Discussion

• The limitations (starts on line 260) should be a separate paragraph.

• Your last paragraph is a single sentence. Expanding this paragraph with a couple more sentences would help highlight the novelty and implications of this study.

**Do you want your identity to be public for this peer review?** For information about this choice, including consent withdrawal, please see our Privacy Policy

Reviewer #1: No

Reviewer #2: No

---

## [Author Response · Author response to Decision Letter 2]

2 May 2025

To the editors:

Thank you for your interest and the opportunity to revise our manuscript. We have responded to the reviewer comments below and uploaded a modified manuscript for consideration. We have also updated manuscript formatting based on the PLOS One style guide.

Reviewer #1

Comment 1: Line 18: I could not understand the meaning of the term "drug use profiles" until I read the Discussion(line 227). I recommend that the authors add some explanation.

AUTHOR RESPONSE 1: We added an explanation for this term in the abstract beginning on line 18.

Comment 2: Line 20: I guess 16� is "non-"ADF ER.

AUTHOR RESPONSE 2: This has been updated to be more clear.

Comment 3: Are decimal points in percentages rounded down? Is it rounded off? It does not seem to be consistent.

AUTHOR RESPONSE 3: All percentages have been rounded to 1 decimal place.

Materials and Methods

Comment 4: Even if the authors wrote about the sponsor of this research in Acknowledgment, I think that it should also be written in Materials and Methods.

AUTHOR RESPONSE 4: Funding disclosure added, beginning on line 137.

Comment 5: Line 106: I understand the classification of comparable products, however, I could not know how the authors chose comparable products. Additionally, the list of products should be shown in a table.

AUTHOR RESPONSE 5: Comparable products were selected based on having similar indication and forms to XTAMPZA ER. This has been clarified beginning on line 122 and a full list of products included in each group has been added as an appendix table.

Comment 6: Lines 30-43: The authors wrote the number of participants. It should also be shown as a flow diagram.

AUTHOR RESPONSE 6: Flow chart was added as figure 1 on line 163.

Comment 7: The questionnaire of this survey should be shown.

AUTHOR RESPONSE 7: Wording of the survey has been included as an appendix.

Result

Comment 8: Table 2: In the "Reason for Tampering", "To increase the high feeling from the pill", "To improve the pain relief from the pill", and "To feel the effects of the pill more quickly" should be separate from other items. I think that "To swallow", "inject", "smoke or vape", and "snort" mean only change the route, do not mean the reasons. These are different concepts.

AUTHOR RESPONSE 8: The reasons described in the manuscript reflect the way the questions were asked on the survey (exact survey language added as supplemental). Because these products are not intended for non-oral use, use by other routes motivates tampering with the pills and that motivation was considered a reason in how the question was developed. This reasoning was added beginning on line 133.

Comment 9: Table 2: In the "Risk Behaviors and Outcomes" should be distinguished from "Risk Behaviors" and "Outcomes".

AUTHOR RESPONSE 9: This table has been updated to separate outcomes from risk behaviors.

Comment 10: Figures should be improved. I can imagine what the authors want to show in the Figures, however, it is unclear on what basis each point is placed in its respective position. I know the two X axes show "mean difficulty" and "mean perceived risk of tampering", respectively. The actual value of the respective means should also be specified somewhere.

AUTHOR RESPONSE 10: Actual means corresponding to each point have been added as captions to both figures. The methods have also been updated to include the numerical values of the Likert scale for better interpretability of the means.

Discussion

Comment 11: Lines 214: The figures 33.8� and 7.7% are not shown in the Result section and Tables. It is unfriendly for readers. I recommend that the authors show these in the Results section.

AUTHOR RESPONSE 11:

Comment 12: The number of consumptions, the number of prescriptions dispensed, or the ratio of the products surveyed in this study in the US is required. If XTAMPZA is not commonly prescribed and distributed in the US, the number of participants who non-medically use XTAMPZA will also be small.

AUTHOR RESPONSE 12: The proportion of dispensing for each comparator group relative to XTAMPZA ER was added beginning on line 154.

Reviewer #2

Comment 1: Please double-check punctions and grammar.

AUTHOR RESPONSE 1: Both coauthors and myself have reviewed the manuscript for punctuation and grammar, making several corrections.

Introduction

Comment 2: There is an extra “.” On line 41.

AUTHOR RESPONSE 2: Extra punctuation removed.

Comment 3: Need to capitalize the start of the sentence on line 74.

AUTHOR RESPONSE 3: “f” changed to “F”.

Comment 4: Missing a period at the end of the sentence on line 76.

AUTHOR RESPONSE 4: punctuation added.

Comment 5: There is an extra “.” and “,” on line 80.

AUTHOR RESPONSE 5: Extra punctuation removed

Comment 6: You explain what NMU stands for in line 95, but it needs to occur at the first usage of NMU (line 85).

AUTHOR RESPONSE 6: definition of NMU abbreviation added to first instance on line 86 and removed from line 96.

Results

Comment 7: This may be an oversight on my end, but I do not see where Figures 1 or 2 are supposed to be placed in the manuscript.

AUTHOR RESPONSE 7: Figure 1 has a title/caption beginning on line 196 and figure 2 has a title/caption beginning on line 212.

Discussion

Comment 8: The limitations (starts on line 260) should be a separate paragraph.

AUTHOR RESPONSE 8: The limitations have been split into a paragraph separate from the strengths, beginning on line 274, with some added content.

Comment 9: Your last paragraph is a single sentence. Expanding this paragraph with a couple more sentences would help highlight the novelty and implications of this study.

AUTHOR RESPONSE 9: The concluding paragraph has been expanding to highlight the novelty and important results of this study.

---

## [Decision Letter · Decision Letter 2]

7 Jul 2025

Dear Dr. Burkett,

Thank you for submitting your manuscript to PLOS ONE. After careful consideration, we feel that it has merit but does not fully meet PLOS ONE’s publication criteria as it currently stands. Therefore, we invite you to submit a revised version of the manuscript that addresses the points raised during the review process.

Based on the revised manuscript and the authors' responses to reviewers, the paper is well-structured, data-rich, and addresses major reviewer concerns. However, a few key areas for further improvement remain that could enhance the manuscript’s clarity, rigor, and reader trust. Some part of the abstract and discussion are still wordy. This may hinder understanding for broader audiences. Perform a final language polish focusing on removing redundant phrases like in line 285-286. Simplifying overly long sentences, in the Abstract (line 18–19) and Discussion (line 287- 289).

We look forward to receiving your revised manuscript.

Kind regards,

Hope Onohuean, PhD

Academic Editor

PLOS ONE

Journal Requirements:

Reviewers' comments:

Reviewer's Responses to Questions

**Comments to the Author**

Reviewer #2: All comments have been addressed

Reviewer #3: All comments have been addressed

2. Is the manuscript technically sound, and do the data support the conclusions?

Reviewer #2: Yes

Reviewer #3: Yes

3. Has the statistical analysis been performed appropriately and rigorously?

Reviewer #2: Yes

Reviewer #3: Yes

4. Have the authors made all data underlying the findings in their manuscript fully available?

Reviewer #2: Yes

Reviewer #3: Yes

5. Is the manuscript presented in an intelligible fashion and written in standard English?

Reviewer #2: Yes

Reviewer #3: Yes

Reviewer #2: All my concerns have been addressed. I applaud the authors for their dedicated efforts to develop this manuscript to its current form.

Reviewer #3: Based on the revised manuscript and the authors' responses to reviewers, the paper is well-structured, data-rich, and addresses major reviewer concerns. However, a few key areas for further improvement remain that could enhance the manuscript’s clarity, rigor, and reader trust. Some part of the abstract and discussion are still wordy. This may hinder understanding for broader audiences. Perform a final language polish focusing on removing redundant phrases like in line 285-286. Simplifying overly long sentences, in the Abstract (line 18–19) and Discussion (line 287- 289).

**Do you want your identity to be public for this peer review?** For information about this choice, including consent withdrawal, please see our Privacy Policy

Reviewer #2: No

Reviewer #3: No

---

## [Author Response · Author response to Decision Letter 3]

20 Aug 2025

To the editors:

Thank you for your interest and the opportunity to revise our manuscript. We have responded to the reviewer comments below and uploaded a modified manuscript for consideration. We have also updated manuscript formatting based on the PLOS One style guide.

Reviewer #3

Comment 1: Reviewer #3: Based on the revised manuscript and the authors' responses to reviewers, the paper is well-structured, data-rich, and addresses major reviewer concerns. However, a few key areas for further improvement remain that could enhance the manuscript’s clarity, rigor, and reader trust. Some part of the abstract and discussion are still wordy. This may hinder understanding for broader audiences. Perform a final language polish focusing on removing redundant phrases like in line 285-286. Simplifying overly long sentences, in the Abstract (line 18–19) and Discussion (line 287- 289).

AUTHOR RESPONSE 1: We performed a thorough round of editing, focusing on removing redundant text and simplifying long sentences.

---

## [Editor Report · Decision Letter 3]

2 Sep 2025

Motivations, perceived risk, and tampering of XTAMPZA ER and abuse-deterrent opioid drugs

PONE-D-24-32479R3

Dear Dr. Burkett,

We’re pleased to inform you that your manuscript has been judged scientifically suitable for publication and will be formally accepted for publication once it meets all outstanding technical requirements.

Kind regards,

Hope Onohuean, PhD

Academic Editor

PLOS ONE
---

## [Editor Report · Acceptance letter]

PONE-D-24-32479R3

PLOS ONE

Dear Dr. Burkett,

I'm pleased to inform you that your manuscript has been deemed suitable for publication in PLOS ONE. Congratulations! Your manuscript is now being handed over to our production team.

Kind regards,

on behalf of

Dr. Hope Onohuean

Academic Editor

PLOS ONE